# DIFFERENTIALLY PRIVATE META-LEARNING

**Jeffrey Li, Mikhail Khodak, Sebastian Caldas**
Carnegie Mellon University
`jwl3@cs.cmu.edu`

**Ameet Talwalkar**
Carnegie Mellon University & Determined AI

## ABSTRACT

Parameter-transfer is a well-known and versatile approach for meta-learning, with applications including few-shot learning, federated learning, and reinforcement learning. However, parameter-transfer algorithms often require sharing models that have been trained on the samples from specific tasks, thus leaving the task-owners susceptible to breaches of privacy. We conduct the first formal study of privacy in this setting and formalize the notion of *task-global differential privacy as a practical relaxation of more commonly studied threat models*. We then propose a new differentially private algorithm for gradient-based parameter transfer that not only satisfies this privacy requirement but also retains provable transfer learning guarantees in convex settings. Empirically, we apply our analysis to the problems of federated learning with personalization and few-shot classification, showing that allowing the relaxation to task-global privacy from the more commonly studied notion of *local privacy* leads to dramatically increased performance in recurrent neural language modeling and image classification.

## 1 INTRODUCTION

The field of *meta-learning* offers promising directions for improving the performance and adaptability of machine learning methods. At a high level, the key assumption leveraged by these approaches is that the *sharing* of knowledge gained from individual learning tasks can help catalyze the learning of similar unseen tasks. However, the collaborative nature of this process, in which task-specific information must be sent to and used by a *meta-learner*, also introduces inherent data privacy risks.

In this work, we focus on a popular and flexible meta-learning approach, *parameter transfer* via *gradient-based meta-learning* (GBML). This set of methods, which includes well-known algorithms such as MAML (Finn et al., 2017) and Reptile (Nichol et al., 2018), tries to learn a common initialization $\phi$ over a set of tasks $t = 1, \ldots, T$ such that a high-performance model can be learned in only a few gradient-steps on new tasks. Notably, information flows constantly between training tasks and the meta-learner as learning progresses; to make iterative updates, the meta-learner obtains feedback on the current $\phi$ by having task-specific models $\bar{\theta}_t$ trained with it.

Meanwhile, in many settings amenable to meta-learning, it is crucial to ensure that sensitive information in each task's dataset stays private. Examples of this include learning models for word prediction on cell phone data (McMahan et al., 2018), clinical predictions using hospital records (Zhang et al., 2019), and fraud detectors for competing credit card companies (Stolfo et al., 1997). In such cases, each data-owner can benefit from information learned from other tasks, but each also desires, or is legally required, to keep their raw data private. Thus, it is not sufficient to learn a well-performing $\phi$; it is equally imperative to ensure that a task's sensitive information is not obtainable by *anyone* else.

While parameter transfer algorithms can move towards this goal by peforming task-specific optimization locally, thus preventing direct access to private data, this provision is far from fail-safe in terms of privacy. A wealth of work has shown in the single-task setting that it is possible for an adversary with only access to the model to learn detailed information about the training set, such as the presence or absence of specific records (Shokri et al., 2017) or the identities of sensitive features given other covariates (Fredrikson et al., 2015). Furthermore, Carlini et al. (2018) showed that deep

neural networks can effectively memorize user-unique training examples, which can be recovered even after only a single epoch of training. As such, in parameter-transfer methods, the meta-learner or any downstream participant can potentially recover data from a previous task.

However, despite these serious risks, privacy-preserving meta-learning has remained largely an unstudied problem. Our work aims to address this issue by applying *differential privacy* (DP), a well-established definition of privacy with rich theoretical guarantees and consistent empirical success at preventing leakages of data (Carlini et al., 2018; Fredrikson et al., 2015; Jayaraman and Evans, 2019). Crucially, although there are various threat models and degrees of DP one could consider in the meta-learning setting (as we outline in Section 2), we balance the well-documented trade-off between privacy and model utility by formalizing and focusing on a setting that we call *task-global* DP. This setting provides a strong privacy guarantee for each task-owner that sharing $\hat{\theta}_t$ with the meta-learner will not reliably reveal anything about specific training examples to *any* downstream agent. It also allows us to use the framework of Khodak et al. (2019a) to provide a DP GBML algorithm that enjoys provable learning guarantees in convex settings.

Finally, we show an application of our work by drawing connections to federated learning (FL) (Li et al., 2019). While standard methods for FL, such as FedAvg (McMahan et al., 2017), have inspired many works also concerning DP in a multi-user setup (Agarwal et al., 2018; Bhowmick et al., 2019; Geyer et al., 2018; McMahan et al., 2018; Truex et al., 2019), we are the first to consider task-global DP as a useful variation on standard DP settings. Moreover, these works fundamentally differ from ours in that they do not consider a task-based notion of learnability, instead focusing on the global federated learning problem to learn a single global model. That being said, a federated setting involving per-user personalization (Chen et al., 2018; Smith et al., 2017) is a natural meta-learning application.

More specifically, our main contributions are:

1. We are the first to provide a taxonomy for the different notions of DP possible for meta-learning. In particular, we formalize on a variant we call *task-global* DP, showing and arguing that it adds a useful option to commonly studied settings in terms of trading privacy and accuracy.
2. We propose the first DP GBML algorithm, which we construct to satisfy this privacy setting. Further, we show a straightforward extension for obtaining a *group DP* version of our setting to protect multiple samples simultaneously.
3. While our privacy guarantees hold generally, we also prove learning-theoretic results in convex settings. Our learning guarantees scale with task-similarity, as measured by the closeness of the task-specific optimal parameters (Denevi et al., 2019; Khodak et al., 2019b).
4. We show that our algorithm, along with its theoretical guarantees, naturally carries over to federated learning with personalization. Compared to previous notions of privacy considered in works for DP federated learning (Agarwal et al., 2018; Bhowmick et al., 2019; Geyer et al., 2018; McMahan et al., 2018; Truex et al., 2019), we are, to the best of our knowledge, the first to simultaneously provide both privacy and learning guarantees.
5. Empirically, we demonstrate that our proposed privacy setting allows for strong performance on federated language-modeling and few-shot image classification tasks. For the former, we achieve close to the performance of non-private models and significantly improve upon the performance of models trained with local-DP guarantees, a previously studied notion that also provides protections against the meta-learner. Our setting reasonably relaxes this latter notion but can achieve roughly 1.7–2.3 times the accuracy on a modified version of the Shakespeare dataset (Caldas et al., 2018) and 1.6–1.7 times the accuracy on a modified version of Wiki-3029 (Arora et al., 2019) across various privacy budgets. For image-classification, we show that we show that we can still retain significant benefits of meta-learning while applying task-global DP on Omniglot (Lake et al., 2011) and Mini-ImageNet (Ravi and Larochelle, 2017).

## 1.1 RELATED WORK

**DP Algorithms in Federated Learning Settings.** Works most similar to ours focus on providing DP for federated learning. Specifically, Geyer et al. (2018) and McMahan et al. (2018) apply update clipping and the Gaussian Mechanism to achieve user-level global DP federated learning algorithms for language modeling and image classification tasks respectively. Their methods are shown to only suffer minor drops in accuracy compared to non-private training but they do not consider protections

to inferences made by the meta-learner. Alternatively, Bhowmick et al. (2019) does achieve such protection by applying a theoretically rate-optimal local DP mechanism on the $\bar{\theta}_t$'s users send to the meta-learner. However, they sidestep hard minimax rates (Duchi et al., 2018) by assuming the central server has limited side-information and allowing for a large privacy budget. In this work, though we achieve a relaxation of the privacy of Bhowmick et al. (2019), we do not restrict the adversary's power. Finally, Truex et al. (2019) does consider a setting that coincides with task-global DP, but they focus primarily on the added benefits of applying MPC (see below) rather than studying the merits of the setting in comparison to other potential settings.

**Secure Multiparty Computation (MPC).** MPC is a cryptographic technique that allows parties to calculate a function of their inputs while also maintaining the privacy of each individual inputs. In GBML, sets of model updates may come in a batch from multiple tasks, and hence MPC can securely aggregate the batch before it is seen by the meta-learner. Though MPC itself gives no DP guarantees against future inference, it can combined with DP to increase privacy. This approach has been studied in the federated setting, e.g. by Agarwal et al. (2018), who apply MPC in the same difficult setting of Bhowmick et al. (2019), and Truex et al. (2019), who apply MPC similarly to a setting analogous to ours. On the other hand, MPC also comes with additional practical challenges such as peer-to-peer communication costs, drop outs, and vulnerability to collaborating participants. As such, combined with its applicability to multiple settings, including ours, we consider MPC to be an orthogonal direction.

## 2 PRIVACY IN A META-LEARNING CONTEXT

In this section, we first formalize the meta-learning setting that we consider. We then describe the various threat models that arise in the GBML setup, before presenting the different DP notions that can be achieved. Finally, we highlight the specific model and type of DP that we analyze.

### 2.1 PARAMETER TRANSFER META-LEARNING

In parameter transfer meta-learning, we assume that there is a set of learning tasks $t = 1, \ldots, T$, each with its corresponding disjoint training set $D_t$. Each $D_t$ contains $m_t$ training examples $\{z_{t,i}\}_{i=1}^{m_t}$ where each $z_{t,i} \in \mathcal{X} \times \mathcal{Y}$. The goal within each task is to learn a function $f_{\hat{\theta}_t} : \mathcal{X} \to \mathcal{Y}$ parameterized by $\hat{\theta}_t \in \Theta \subset \mathbb{R}^d$ that performs "well," generally in the sense that it has low within-task population risk in the distributional setting. The meta-learner's goal is to learn an initialization $\phi \in \Theta$ that leads to a well-performing $\hat{\theta}_t$ within-task. In GBML this $\phi$ is learned via an iterative process that alternates between the following two steps: (1) a within-task procedure where a batch of task-owners $B$ receives the current $\phi$ and each $t \in B$ uses $\phi$ as an initialization for running a within-task optimization procedure, obtaining $\bar{\theta}_t(D_t, \phi)$; (2) a meta-level procedure where the meta-learner receives these model updates $\{\bar{\theta}_t\}_{t \in B}$ and aggregates them to determine an updated $\phi$. Note that we do not assume $\hat{\theta}_t = \bar{\theta}_t$, as the updates shared for the meta-learning procedure can be obtained from a different procedure than the refined model used for downstream within-task inference. This is especially the case when concerning the addition of noise for DP as part of the meta-learning procedure.

### 2.2 THREAT MODELS FOR GBML

As in any privacy endeavor, before discussing particular mechanisms, a key specification must be made in terms of what threat model is being considered. In particular, it must be specified both (1) who the potential adversaries are and (2) what information needs to be protected.

**Potential adversaries.** For a single task-owner, adversaries may be either solely recipients of $\phi$ (i.e. other task-owners) or recipients of either $\phi$ or $\bar{\theta}_t$ (i.e. also the meta-learner). In the latter case, we consider only a honest-but-curious meta-learner, who does not deviate from the agreed upon algorithm but may try to make inferences from $\bar{\theta}_t$. In both cases, concern is placed not only about these other participants' intentions, but also their own security against access by malicious outsiders.

**Data to be protected.** A system can choose either to protect information contained in single records $z_{t,i}$ one-at-a-time or to protect entire datasets $D_t$ simultaneously. This distinction between *record-level* and *task-level* privacy can be practically important. Multiple $z_{t,i}$ within $D_t$ may reveal the same secret (e.g., a cell-phone user has sent their SSN multiple times), or the entire distribution

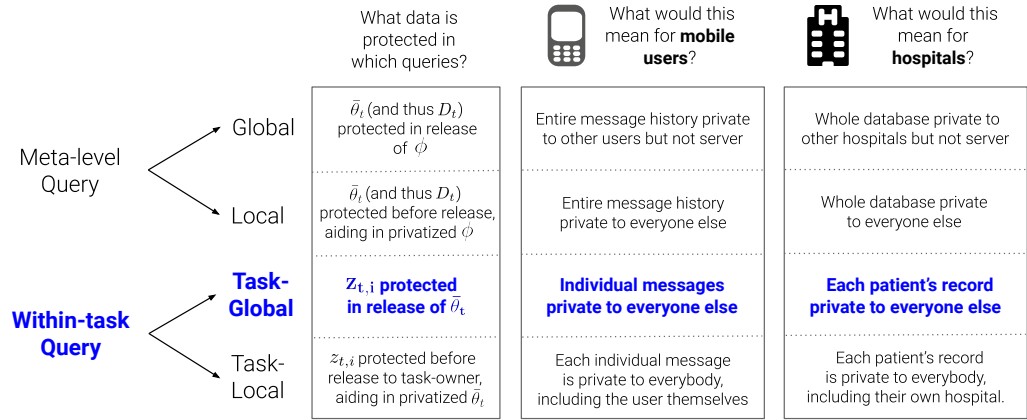

Figure 1: Summary of the privacy protections guaranteed by local and global DP at the different levels of the meta-learning problem (with our notion in blue). On the right, we show what each specification would mean in two practical federated scenarios: mobile users and hospital networks.

of $D_t$ could reveal sensitive information (e.g., a user has sent all messages in a foreign language). In these cases, record-level privacy may not be sufficient. However, given that privacy and utility are often at odds, we often seek the weakest notion of privacy needed in order to best preserve utility.

In related work, focus has primarily been placed on task-level protections. However, such approaches usually fall into two extremes, either obtaining strong learning but having to trust the meta-learner (McMahan et al., 2018; Geyer et al., 2018) or trusting nobody but also obtaining low performance (Bhowmick et al., 2019). In response, we try to bridge the gap between these threat models by considering a model that makes a relaxation from task-level to record-level privacy but retains protections for each task-owner against *all* other parties. This relaxation can be reasonably justified in practical situations, as while task-level guarantees are strictly stronger, they may also be unnecessary. In particular, record-level guarantees are likely to be sufficient whenever single records each pertain to different individuals. For example, for hospitals, what we care about is providing privacy to the individual patients and not aggregate hospital information. For cell-phones, if one can bound the number of texts that could reveal the *same* sensitive information, then a straightforward extension of our setting and methods, which protects up to $k$ records simultaneously, could also be sufficient.

## 2.3 DIFFERENTIAL PRIVACY (DP) IN A SINGLE-TASK SETTING

In terms of actually achieving privacy guarantees for machine learning, a de-facto standard has been to apply DP, a provision which strongly limits what one can infer about the examples a given model was trained on. Assuming a training set $D = \{z_1, \ldots, z_m\}$, two common types of DP are considered.

**Differential Privacy (Global DP).** A randomized mechanism $\mathcal{M}$ is $(\varepsilon, \delta)$-*differentially private* if for all measurable $\mathcal{S} \subseteq \text{Range}(\mathcal{M})$ and for all datasets $D, D'$ that differ by at most one element:

$$\mathbb{P}[\mathcal{M}(D) \in \mathcal{S}] \leq e^{\varepsilon} \mathbb{P}[\mathcal{M}(D') \in \mathcal{S}] + \delta$$

If this holds for $D, D'$ differing by at most $k$ elements, then $(\varepsilon, \delta)$ *k-group DP* is achieved.

**Local Differential Privacy.** A randomized mechanism $\mathcal{M}$ is $(\varepsilon, \delta)$-*locally differentially private* if for any two possible training examples $z, z' \in \mathcal{X} \times \mathcal{Y}$ and measurable $\mathcal{S} \subseteq \mathcal{X} \times \mathcal{Y}$:

$$\mathbb{P}[\mathcal{M}(z) \in \mathcal{S}] \leq e^{\varepsilon} \mathbb{P}[\mathcal{M}(z') \in \mathcal{S}] + \delta$$

Global DP guarantees the difficulty of inferring the presence of a specific record in the training set by observing $\mathcal{M}(D)$. It assumes a trusted *aggregator* running $\mathcal{M}$ gets direct access to $D$ and privatizes the final output. Meanwhile, local DP assumes more strictly that the aggregator also cannot be trusted, thus requiring a random mechanism to be applied individually on each $z$ *before* training. However, it generally results in worse model performance, suffering from hard minimax rates (Duchi et al., 2018).

Table 1: Broad categorization of the DP settings considered by our work in meta-learning and notable past works in the federated setting. Note that by using a de-centralized method for aggregation, Agarwal et al. (2018) can still protect against the meta-learner from making inferences on any individual $\hat{\theta}_t$ with what is only effectively a global DP mechanism.

| Previous Work | Notion of DP | Privacy for $\phi$ | Privacy for $\bar{\theta}_t$ |
|---|---|---|---|
| McMahan et al. (2018) | Global | Task-level | - |
| Geyer et al. (2018) | Global | Task-level | - |
| Bhowmick et al. (2019) | Local, Global | Task-level | Task-level |
| Agarwal et al. (2018) | Global + MPC | Task-level | Task-level |
| Truex et al. (2019) | Task-Global + MPC | Record-level | Record-level |
| Our work | Task-Global | Record-level | Record-level |

## 2.4 DIFFERENTIAL PRIVACY FOR A GBML SETTING

In meta-learning, there exists a hierarchy of agents and statistical queries, so we cannot as simply define global and local DP. Here, both the meta-level sub-procedure, $\{\bar{\theta}_t\}_{t \in B} \to \phi$, and the within-task sub-procedure, $\{z_{t,i}\}_{i=1}^{m_t} \to \bar{\theta}_t$, can be considered individual queries and a DP algorithm can implement either to be DP. Further, for each query, the procedure may be altered to satisfy either local DP or global DP. Thus, there are four fundamental options that follow from standard DP definitions.

(1) *Global DP:* Releasing $\phi$ will at no point compromise information regarding any specific $\bar{\theta}_t$.
(2) *Local DP:* Additionally, each $\bar{\theta}_t$ is protected from being revealed to the meta-learner.
(3) *Task-Global DP:* Releasing $\bar{\theta}_t$ will at no point compromise any specific $z_{t,i}$.
(4) *Task-Local DP:* Additionally, each $z_{t,i}$ is protected from being revealed to task-owner.

To form analogies to single-task DP, the examples in the meta-level procedure are the model updates and the aggregator is the meta-learner. For the within-task procedure, the examples are actually the individual records and the aggregator is the task-owner. As such, (1) is implemented by the meta-learner, (2) and (3) are implemented by the task-owner, and (4) is implemented by record-owners. By immunity to post-processing, the guarantees for (3) and (4) also automatically apply to the release of any future iteration of $\phi$, thus protecting against future task-owners as well. Meanwhile, though (1) and (2) by definition protect the identities of individual $\bar{\theta}_t$, they actually satisfy a task-level threat model by doing so. Intuitively, not being able to reliably infer anything about $\bar{\theta}_t$ implies that nothing can be inferred about the $D_t$ that was used to generate it.

Using the terminology we introduce in Section 2.4, previous works for DP in federated settings can be categorized as in Table 1. While these works do not assume a multi-task setting, we can still naturally use the terms *global/local* and *task-global/task-local* to analogously refer to releasing the global model (by the central server in the case without MPC) and user-specific updates (by users' devices) respectively.

## 3 DIFFERENTIALLY PRIVATE PARAMETER-TRANSFER

### 3.1 ALGORITHM

We now present our DP GBML method, which is written out in its online (regret) form in Algorithm 1. Here, we observe that both within-task optimization and meta-optimization are done using some form of gradient descent. The key difference between this algorithm and traditional GBML is that since task-learners must send back privatized model updates, each now applies an DP gradient descent procedure to learn $\bar{\theta}_t$ when called. However, at meta-test time the task-learner will run a *non-private* descent algorithm to obtain the parameter $\hat{\theta}_t$ used for inference, as this parameter may remain locally. To obtain learning-theoretic guarantees, we use a variant of Algorithm 1 in which the DP algorithm is an SGD procedure (Bassily et al., 2019, Algorithm 1) that adds a properly scaled Gaussian noise vector at each iteration.

---

**Algorithm 1:** Online version of our $(\varepsilon, \delta)$-meta-private parameter-transfer algorithm.

---

Meta-learner picks first meta-initialization $\phi_1 \in \Theta$.
**for** task $t \in [T]$ **do**

  Meta-learner sends meta-initialization $\phi_t$ to task $t$.
  Task-learner runs OGD starting from $\theta_{t,1} = \phi_t$ on losses $\{\ell_{t,i}\}_{i=1}^m$ to obtain $\hat{\theta}_t$.
  Task-learner $t$ runs $(\varepsilon, \delta)$-DP algorithm (noisy-SGD) on losses $\{\ell_{t,i}\}_{i=1}^m$ to get $\bar{\theta}_t$.
  Task-learner sends $\bar{\theta}_t$ to meta-learner.
  Meta-learner constructs loss $\ell_t(\phi) = \frac{1}{2}\|\bar{\theta}_t - \phi_t\|_2^2$.
  Meta-learner updates meta-initialization $\phi_{t+1}$ using an OCO algorithm on $\ell_1, \ldots, \ell_t$.

**Result:** Meta-initialization $\hat{\phi} = \frac{1}{T}\sum_{t=1}^T \phi_t$ to use on test tasks.

---

## 3.2 PRIVACY GUARANTEES

We run a certified $(\varepsilon, \delta)$-DP version of SGD (Bassily et al., 2019, Algorithm 1) within each task. Therefore, this guarantees that the contribution of each task-owner, a $\bar{\theta}_t$ trained on their data, carries global DP guarantees with respect to the meta-learner. Additionally, since DP is preserved under post-processing, the release of any future calculation stemming from $\bar{\theta}_t$ also carries the same DP guarantee.

## 3.3 LEARNING GUARANTEES

Our learning result follows the setup of Baxter (2000), who formalized the LTL problem as using task-distribution samples $\mathcal{P}_1, \ldots, \mathcal{P}_T \sim \mathcal{Q}$ from some meta-distribution $\mathcal{Q}$ and samples indexed by $i = 1, \ldots, m$ from those tasks to improve performance when a new task $\mathcal{P}$ is sampled from $\mathcal{Q}$ and we draw $m$ samples from it. In the setting of parameter-transfer meta-learning we are learning functions parameterized by real-valued vectors $\theta \in \Theta \subset \mathbb{R}^d$, so our goal will follow that of Denevi et al. (2019) and Khodak et al. (2019b) in seeking bounds on the transfer-risk – the distributional performance of a learned parameter on a new task from $\mathcal{Q}$ – that improve with task similarity.

The specific task-similarity metric we consider is the average deviation of the risk-minimizing parameters of tasks sampled from the distribution $\mathcal{Q}$ are close together. This will be measured in-terms of the following quantity: $V^2 = \min_{\phi \in \Theta} \frac{1}{2}\mathbb{E}_{\mathcal{P} \sim \mathcal{Q}}\|\theta_{\mathcal{P}} - \phi\|_2^2$, for $\theta_P \in \arg\min_{\theta \in \Theta} \ell_{\mathcal{P}}(\theta)$ a risk-minimizer of task-distribution $\mathcal{P}$. This quantity is roughly the variance of risk-minimizing task-parameters and is a standard quantifier of improvement due to meta-learning (Denevi et al., 2019; Khodak et al., 2019b). For example, Denevi et al. (2019) show excess transfer-risk guarantees of the form $\mathcal{O}\left(\frac{V}{\sqrt{m}} + \sqrt{\frac{\log T}{T}}\right)$ when $T$ tasks with $m$ samples are drawn from the distribution. This guarantee ensures that as we see more tasks our transfer risk becomes roughly $V/\sqrt{m}$, which if the tasks are similar, i.e. $V$ is small, implies that LTL improves over single-task learning.

In Algorithm 1, each user $t$ obtains a within-task parameter $\bar{\theta}_t$ by running (non-private) OGD on a sequence of losses $\ell_{t,1}, \ldots, \ell_{t,m}$ and averaging the iterates. The regret of this procedure, when averaged across the users, implies a bound on the expected excess transfer risk of new task from $\mathcal{Q}$ when running OGD from a learned initialization (Cesa-Bianchi et al., 2004). Thus our goal is to bound this regret in terms of $V$; here we follow the Average Regret-Upper-Bound Analysis (ARUBA) framework of Khodak et al. (2019b) and treat meta-update procedure itself as an online algorithm optimizing a bound on the performance measure (regret) of each within-task algorithm. As OGD's regret depends on the squared distance $\frac{1}{2}\|\theta_t^* - \phi_t\|_2^2$ of the optimal parameter from the initialization $\phi_t$, with no privacy concerns one could simply update $\phi_t$ using $\theta_t^* \in \arg\min_{\theta \in \Theta} \sum_{i=1}^m \ell_{t,i}(\theta)$ to recover guarantees similar to those in Denevi et al. (2019) and Khodak et al. (2019b).

However, this approach requires sending $\theta_t^*$ to the meta-learner, which is not private; instead in Algorithm 1 we send $\hat{\theta}_t$, which is the output of noisy SGD. To apply ARUBA, we need an additional assumption – that the losses satisfy the following quadratic growth (QG) property: for some $\alpha > 0$,

$$\frac{\alpha}{2}\|\theta - \theta_{\mathcal{P}}\|_2^2 \leq \ell_{\mathcal{P}}(\theta) - \ell_{\mathcal{P}}(\theta_{\mathcal{P}}) \quad \forall \theta \in \Theta \tag{1}$$

Here $\theta_{\mathcal{P}}$ is the risk minimizer of $\ell_{\mathcal{P}}$. This assumption, which Khodak et al. (2019a) shows is reasonable in settings such as logistic regression, amounts to a statistical non-degeneracy assumption on the parameter-space – that parameters far away from the risk-minimizer do not have low-risk. Note that assuming the population risk is QG is significantly weaker than assuming strong convexity of the empirical risk, which previous work (Finn et al., 2019) has assumed to hold for task losses but does not hold for applicable cases such as few-shot least-squares or logistic regression if the number of task-samples is smaller than the data-dimension.

We are now able to state our main theoretical result, a proof of which is given in Appendix A. The result follows from a bound on the task-averaged regret across all tasks of a simple online meta-learning procedure that treats the update $\bar{\theta}_t$ sent by each task as an approximation of the optimal parameter in hindsight $\theta_t^*$. Since this parameter determines regret on that task, by reducing the meta-update procedure to OCO on this sequence of functions in a manner similar to (Khodak et al., 2019a), we are able to show a task-similarity-dependent bound. Following this the statistical guarantee stems from a nested online-to-batch conversion, a standard procedure to convert low-regret online-learning algorithms to low-risk distribution-learning algorithms (Cesa-Bianchi et al., 2004).

**Theorem 3.1.** *Suppose $\mathcal{Q}$ is a distribution over task-distributions $\mathcal{P}$ over $G$-Lipschtz, $\beta$-strongly-smooth, 1-bounded convex loss functions $\ell : \Theta \mapsto \mathbb{R}$ over parameter space $\Theta$ with diameter $D$ for $\beta \leq \frac{G}{D} \min\left(\sqrt{\frac{m}{2}}, \frac{\varepsilon m}{2\sqrt{2d\log\frac{1}{\delta}}}\right)$, and let each $\mathcal{P}$ satisfy the quadratic growth property (1). Suppose the distribution $\mathcal{P}_t$ of each task is sampled i.i.d. from $\mathcal{Q}$ and we run Algorithm 1 with the $(\varepsilon, \delta)$-DP procedure of Bassily et al. (2019, Algorithm 1) to obtain $\bar{\theta}_t$ as the average iterate for the meta-update step, using $n \geq 1$ steps and learning rate $\frac{\gamma}{G\sqrt{n}}$ for $\gamma > 0$. Letting $V^2 = \min_{\phi\in\Theta}\frac{1}{2}\mathbb{E}_{\mathcal{P}\sim\mathcal{Q}}\|\theta_{\mathcal{P}} - \phi\|_2^2$, there exist settings of $n, \gamma, \eta$ such that we have the following bound on the expected transfer risk when a new task $\mathcal{P}$ is sampled from $\mathcal{Q}$, $m$ samples are drawn i.i.d. from $\mathcal{P}$, and we run OGD with learning rate $\eta$ starting from $\hat{\phi} = \frac{1}{T}\sum_{t=1}^{T}\phi_t$ and use the average $\hat{\theta}$ of the resulting iterates as the learned parameter:*

$$\mathbb{E}_{\mathcal{P}\sim\mathcal{Q}}\mathbb{E}_{\ell\sim\mathcal{P}}\ell(\hat{\theta}) \leq \mathbb{E}_{\mathcal{P}\sim\mathcal{Q}}\mathbb{E}_{\ell\sim\mathcal{P}}\ell(\theta^*) + \tilde{\mathcal{O}}\left(\frac{V}{\sqrt{m}} + \frac{\alpha D^2}{T} + \frac{1}{\alpha}\max\left(\frac{d\log\frac{1}{\delta}}{\varepsilon^2 m^2}, \frac{1}{m}\right)\right)$$

*Here $\theta^*$ is any element of $\Theta$ and the outer expectation is taken over $\ell_{t,i} \sim \mathcal{P}_t \sim \mathcal{Q}$ and the randomness of the within-task DP mechanism. Note that this procedure is $(\varepsilon, \delta)$-DP.*

Theorem 3.1 shows that one can usefully run a DP-algorithm as the within-task method in meta-learning and still obtain improvement due to task-similarity. Specifically, the standard term of $1/\sqrt{m}$ is multiplied by $V$, which is small if the tasks are related via the closeness of their risk minimizers. Thus we can use meta-learning to improve within-task performance relative to single-task learning. We also obtain a very fast convergence of $\tilde{\mathcal{O}}(1/T)$ in the number of tasks. However, we do gain some $O(1/m)$ terms due to the quadratic growth approximation and the privacy mechanism. Note that the assumption that both the functions and its gradients are Lipschitz-continuous are standard and required by the noisy SGD procedure of Bassily et al. (2019).

This theorem also gives us a relatively straightforward extension if the desire is to provide $(\varepsilon, \delta)$-group-DP. Since any privacy mechanism that provides $(\varepsilon, \delta)$-DP also provides $(k\varepsilon, ke^{(k-1)\epsilon}\delta)$-DP guarantees for groups of size $k$ (Dwork and Roth, 2014), we immediately have the following corollary.

**Corollary 3.1.** *Under the same assumptions and setting as Theorem 3.1, achieving $(\varepsilon, \delta)$-group DP is possible with the same guarantee except replacing $\frac{d\log\frac{1}{\delta}}{\varepsilon^2}$ with $\frac{k^3 d}{\varepsilon} + \frac{k^2 d}{\varepsilon}\left[\frac{1}{\varepsilon}\log\frac{k}{\delta} - 1\right]$*

For constant $k$, this allows us to enjoy the stronger guarantee while maintaining largely the same learning rates. This is a useful result given that in some settings, it may be desired to simultaneously protect small groups of size $k << m_t$, such as protecting entire families for hospital records.

## 4 EMPIRICAL RESULTS

We present results that show it is possible to learn useful deep models in federated scenarios while still preserving privacy against all other participants. Specifically, we evaluate the performance of models that have been trained with a *task-global* DP algorithm in comparison to models that

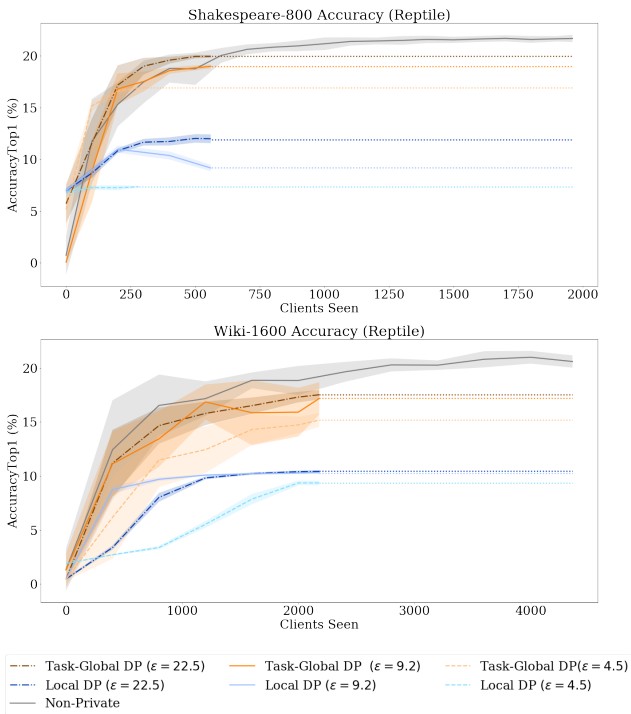

Figure 2: Performance of different versions of Reptile on a next-word-prediction task for two federated datasets. We report the test accuracy on unseen tasks and repeat each experiment 10 times. Solid lines correspond to means, colored bands indicate 1 standard deviation, and dotted lines are for comparing final accuracies (private algorithms can only be trained until privacy budget is met).

have been trained both non-privately and with *local* DP algorithms. We evaluate performance on federated language modeling and few-shot image classification, applying a practical batched variant of Algorithm 1.

**Datasets:**   We train a LSTM-RNN for next word prediction on two federated datasets: (1) The Shakespeare dataset as preprocessed by (Caldas et al., 2018), and (2) a dataset constructed from $3,000$ Wikipedia articles drawn from the Wiki-3029 dataset (Arora et al., 2019), where each article is used as a different task. For each dataset, we set a fixed number of tokens per task, discard tasks with fewer tokens than the specified, and discard samples from those tasks with more. We set the number of tokens per task to $800$ for Shakespeare and to $1,600$ for Wikipedia, divide tokens into sequences of length $10$, and we refer to these modified datasets as Shakespeare-800 and Wiki-1600.

For few-shot image classification, we use the Omniglot (Lake et al., 2011) and Mini-ImageNet (Ravi and Larochelle, 2017) datasets, both with 5-shot-5-way test tasks. As has been done for non-private Reptile (Nichol et al., 2018), we use more training shots at meta-training (trying $m = 10, 20, 30$ for Omniglot and $m = 15, 30, 45$ for Mini-ImageNet) than at meta-test time. Though tasks could be sampled indefinitely, we set a fixed budget of tasks at $T = 10^6$ to reflect to a setting in which a finite number of training tasks constrains our learning and privacy trade-off. For both local and task-global DP, the more tasks that can be grouped in a meta-batch means that less total noise can be added at each round. However, this also means fewer iterations can be taken. We note that for the smallest values of $m$, this setting is enough for non-private training to essentially achieve the reported final accuracies from Nichol et al. (2018).

**Meta Learning Algorithm.**   We study the performance of our method when applied to the batched version of Reptile (Nichol et al., 2018) (which, in our setup, reduces to personalized Federated Averaging when the meta-learning rate is set to $1.0$). For the language modelling tasks, we tune various configurations of task batch size for all methods. We also allow for multiple visits per client, though at the cost of more added noise per iteration for the private methods. Additionally,

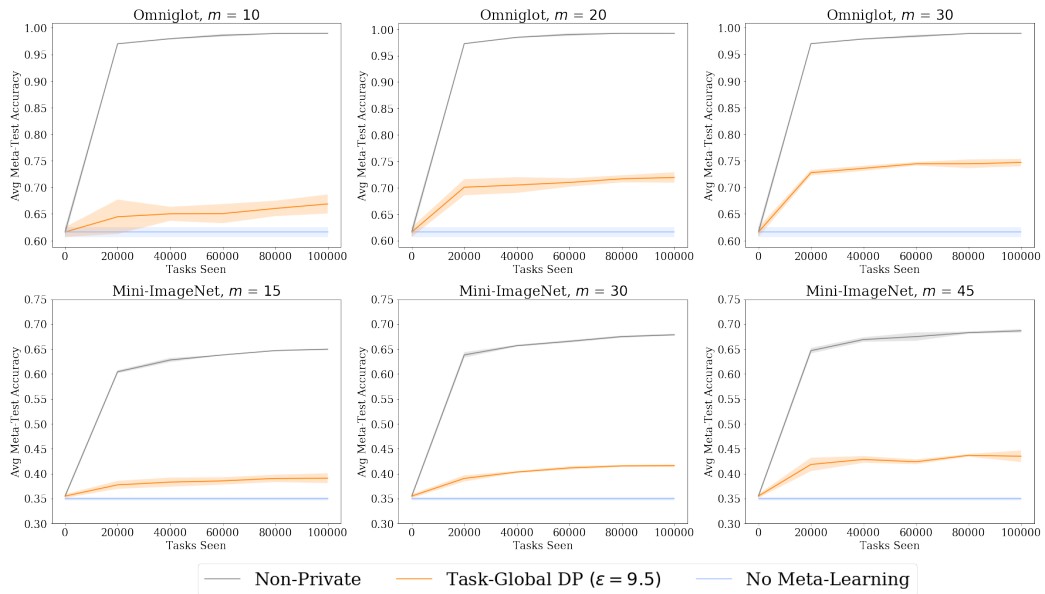

Figure 3: Performance of task-global DP Reptile on 5-shot-5-way Omniglot and Mini-ImageNet. $10^5$ sampled test- tasks were used for evaluation and experiments were repeated 3 times. We do not show a line for Local DP since all hyperparameter settings tried for Local DP resulted in worse performance than the "No Meta-Learning" baseline, whose performance can always be recovered.

for language modeling, we implement gradient clipping and exponential decay on the meta learning rate. For Omniglot and Mini-ImageNet, we use largely the same parameters as Nichol et al. (2018) but we tune the parameters most directly related to privacy: the $L_2$ clipping threshold, the Adam Learning Rate at meta-training time, the meta-batch size, and the within-task batch size. We defer a more complete discussion of hyperparameter tuning to Appendix B.

**Privacy Considerations.** For the *task-global* DP models, we set $\delta = 10^{-3} < \frac{1}{m^{1.1}}$ by convention on each task and we implement DP-SGD (for language modelling) and DP-Adam (for image-classification) within-task using the tools provided by *TensorFlow Privacy*[1], using the *RDP accountant* to track our privacy budgets. Although these algorithms differ from the one presented in Section 3, they still let us realistically explore the efficacy of considering *task-global* privacy. For the language modeling datasets, we try three different privacy budgets (as determined relative to each other by successively doubling the amount of noise added when the goal is to take 1 full gradient step per task) and make sure that all training tasks are sampled without replacement with a fixed batch size until all are seen. This is necessary since multiple visits to a single client results in degradation of the privacy guarantee for that client. We instead aim to provide the same guarantee for each client. For local-DP, though this notion of DP is stronger, we explore the same privacy budgets so as to obtain guarantees that are of the same *confidence*. Here, we essentially run the DP-FedAvg algorithm from (McMahan et al., 2018) with some key changes. First, to get local DP instead of global, we add Gaussian noise to each clipped set of model updates before returning them to the central server instead of after aggregation. Second, while additional gradient steps within-task do not increase the amount of noise required, we do again iterate through tasks without replacement. Unlike for global DP, we cannot hope to have any privacy boosts due to sub-sampling if the meta-learner knows who it is communicating with.

**Results.** Figure 2 shows the performance of both the non-private and *task-global* private versions of Reptile (Nichol et al., 2018) for the language modelling tasks across three different privacy budgets. As expected, neither private algorithm reaches the same accuracy of the non-private version of the algorithm. Nonetheless, the task-global version still comes within $78\%, 88\%,$ and $92\%$ of the non-private accuracy for Shakespeare-800 and within $72\%, 82\%,$ and $83\%$ for Wiki-1600. Meanwhile

---

[1]https://github.com/tensorflow/privacy

achieving local DP results in only about $55\%$ and $50\%$ of the non-private accuracy on both datasets for the *most* generous privacy budget. In practice, these differences can be toggled by further changing the privacy budget or continuing to trade off more training iterations for larger noise multipliers.

We display results for few-shot image classification on Omniglot and Mini-ImageNet in Figure 3. In this setting, *not* applying meta-learning results in meta-test accuracies of around $62\%$ and $36\%$, respectively. Thus, while performance is indeed lower than non-private learning, applying task-global DP *does* result in meta-learning benefits for test-time tasks. In settings where privacy is a concern, this increase in performance is still significantly advantageous for the "task-owners"– test-time tasks (who hold less data). On average, they are able to obtain better models and are still guaranteed privacy at a single-digit $\varepsilon$. Intuitively, larger training-task datasets make it easier to apply privacy within-task, and in accordance with our learning guarantees, adding training shots indeed closes the gap in performance between task-global DP Reptile and non-private Reptile. In comparison, applying local-DP for a similar hyperparameter range consistently decreases performance at test-time. However, the no-meta-learning baseline is a theoretical lower bound for local-DP, as one could set the clipping threshold or meta-learning rate close to $0$ to recover the effects of no meta-learning.

## 5  Conclusions

In this work, we have outlined and studied the issue of privacy in the context of meta-learning. Focusing on the class of gradient-based parameter-transfer methods, we used differential privacy to address the privacy risks posed to task-owners by sharing task-specific models with a central meta-learner. To do so, we formalized and considered the notion of *task-global* differential privacy, which guarantees that individual examples from the tasks are protected from all downstream agents (and particularly the meta-learner). Working in this privacy model, we developed a differentially private algorithm that guarantees both this protection as well as learning-theoretic results in the convex setting. Finally, we demonstrate how this notion of privacy can translate into useful deep learning models for non-convex language modelling and image-classification tasks.

## Acknowledgments

This work was supported in part by DARPA FA875017C0141, the National Science Foundation grants IIS1618714, IIS1705121, and IIS1838017, an Okawa Grant, a Google Faculty Award, an Amazon Web Services Award, a JP Morgan A.I. Research Faculty Award, and a Carnegie Bosch Institute Research Award. Any opinions, findings and conclusions or recommendations expressed in this material are those of the author(s) and do not necessarily reflect the views of DARPA, the National Science Foundation, or any other funding agency.

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

## A    PROOFS OF LEARNING GUARANTEES

**Setting A.1.** *We assume that at each time-step $t$ an adversary chooses a task-distribution $\mathcal{P}_t$ over loss-functions on $\Theta \subset \mathbb{R}^d$ and samples $m$ loss functions $\ell_{t,i}$ for $i \in [m]$. At each time-step $t$ the task-learner receives a parameter $\phi_t$ from the meta-learner, runs online gradient descent with step-size $\eta > 0$ starting from $\phi_t$, and uses the average iterate $\hat{\theta}_t$ as its learned parameter. The task-learner also runs Algorithm 1 of Bassily et al. (2019) for $n = \min\left\{\frac{m}{8}, \frac{\varepsilon^2 m^2}{32d \log\frac{1}{\delta}}\right\}$ steps with learning rate $\frac{\gamma}{G\sqrt{n}} > 0$ on these loss functions and sends the result $\bar{\theta}_t$ to the meta-learner. The meta-learner updates $\phi_{t+1} = (1 - 1/t)\phi_t + \bar{\theta}_t/t$. We assume all loss functions are $G$-Lipschitz w.r.t. $\|\cdot\|_2$ and $\beta$-strongly-smooth w.r.t. $\|\cdot\|_2$ for some $\beta \leq \frac{G}{D}\min\left\{\sqrt{\frac{m}{2}}, \frac{\varepsilon n}{2\sqrt{2d\log\frac{1}{\delta}}}\right\}$, where $D$ is the diameter of $\Theta$. For each distribution $\mathcal{P}_t$ let $\ell_t(\theta) = \mathbb{E}_{\ell \sim \mathcal{P}_t}(\theta)$ be its population risk, $\hat{\ell}_t(\theta) = \frac{1}{m}\sum_{i=1}^m \ell_{t,i}(\theta)$ be its empirical risk, and $\theta_t^* \in \arg\min_{\theta \in \Theta} \ell_t(\theta)$ be the closest population risk minimizer to $\bar{\theta}_t$.*

**Lemma A.1.** *In Setting A.1 we have*

$$\mathbb{E}\,\ell_t(\hat{\theta}_t) - \ell_t(\theta_t^*) \leq 5G\left(\frac{\|\phi_t - \theta_t^*\|_2^2}{\gamma} + \gamma\right)\max\left\{\frac{\sqrt{d\log\frac{1}{\delta}}}{\varepsilon m}, \frac{1}{\sqrt{m}}\right\}$$

*Proof.* Similarly to Lemma 3.3 in Bassily et al. (2019), applying standard OGD analysis (e.g. Lemmas 14.1 and 14.9 of Shalev-Shwartz and Ben-David (2014)) to noisy gradient vectors and taking expectations yields

$$\mathbb{E}\left(\hat{\ell}_t(\hat{\theta}_t) - \hat{\ell}_t(\theta_t^*)\right) \leq \frac{\|\phi_t - \theta_t^*\|_2^2}{2\eta n} + \frac{\eta G^2}{2} + \eta \sigma^2 d$$

where $n$ is the number of steps in noisy SGD and $\sigma^2$ is the variance of the noise added at each step. As in the proof of Theorem 3.2 of Bassily et al. (2019), substituting $\sigma^2 = \frac{8nG^2 \log\frac{1}{\delta}}{m^2 \varepsilon^2}$ and applying the stability result in Lemma 3.4 of the same paper yields

$$\mathbb{E}\,\ell_t(\hat{\theta}_t) - \ell_t(\theta_t^*) \leq \frac{\|\phi_t - \theta_t^*\|_2^2}{2\eta n} + \frac{\eta G^2}{2}\left(\frac{16nd \log\frac{1}{\delta}}{m^2 \varepsilon^2} + 1\right) + \frac{\eta G^2 n}{m}$$

Substituting $n = \min\left\{\frac{m}{8}, \frac{\varepsilon^2 m^2}{32d \log\frac{1}{\delta}}\right\}$ and $\eta = \frac{\gamma}{G\sqrt{n}}$ yields the result.    $\square$

**Lemma A.2.** *In Setting A.1, fix some $\phi^* \in \Theta$ and define $\bar{V}^2 = \frac{1}{T} \sum_{t=1}^{T} \mathbb{E} \|\phi^* - \theta_t^*\|_2^2$. Then for $\gamma = \frac{120G}{\alpha} \max \left\{ \frac{\sqrt{d \log \frac{1}{\delta}}}{\varepsilon m}, \frac{1}{\sqrt{m}} \right\}$ we have*

$$\mathbb{E} \sum_{t=1}^{T} \frac{\|\phi_t - \theta_t^*\|_2^2}{2\eta m} \leq \frac{D^2(1 + \log T) + 4\hat{V}^2 T}{2\eta m} + \frac{7200G^2}{\alpha^2 \eta m} \max \left\{ \frac{d \log \frac{1}{\delta}}{\varepsilon^2 m^2}, \frac{1}{m} \right\} T$$

*Proof.* We first bound the left-hand side without the denominator as

$$\mathbb{E} \sum_{t=1}^{T} \|\phi_t - \theta_t^*\|_2^2$$

$$\leq 2 \mathbb{E} \sum_{t=1}^{T} \|\phi_t - \bar{\theta}_t\|_2^2 + \|\bar{\theta}_t - \theta_t^*\|_2^2$$

$$\leq D^2(1 + \log T) + 2 \mathbb{E} \sum_{t=1}^{T} \|\phi^* - \bar{\theta}_t\|_2^2 + \|\bar{\theta}_t - \theta_t^*\|_2^2$$

$$\leq D^2(1 + \log T) + 2 \mathbb{E} \sum_{t=1}^{T} 2\|\phi^* - \theta_t^*\|_2^2 + 3\|\theta_t^* - \bar{\theta}_t\|_2^2$$

$$= D^2(1 + \log T) + 4\bar{V}^2 T + 6 \mathbb{E} \sum_{t=1}^{T} \|\bar{\theta}_t - \theta_t^*\|_2^2$$

$$\leq D^2(1 + \log T) + 4\bar{V}^2 T + \frac{12}{\alpha} \sum_{t=1}^{T} \mathbb{E} \ell_t(\bar{\theta}_t) - \ell_t(\theta_t^*)$$

$$\leq D^2(1 + \log T) + 4\bar{V}^2 T + \frac{60G}{\alpha} \max \left\{ \frac{\sqrt{d \log \frac{1}{\delta}}}{\varepsilon m}, \frac{1}{\sqrt{m}} \right\} \sum_{t=1}^{T} \mathbb{E} \left( \frac{\|\phi_t - \theta_t^*\|_2^2}{\gamma} + \gamma \right)$$

where in the last step we applied Lemma A.1. Substituting $\gamma = \frac{120G}{\alpha} \max \left\{ \frac{\sqrt{d \log \frac{1}{\delta}}}{\varepsilon m}, \frac{1}{\sqrt{m}} \right\}$ yields

$$\mathbb{E} \sum_{t=1}^{T} \|\phi_t - \theta_t^*\|_2^2 \leq D^2(1 + \log T) + 4\bar{V}^2 T + \frac{14400G^2}{\alpha^2} \max \left\{ \frac{d \log \frac{1}{\delta}}{\varepsilon^2 m^2}, \frac{1}{m} \right\} T$$

The result follows by dividing by $2\eta m$. $\qquad \square$

**Theorem A.1.** *In Setting A.1, suppose all distributions $\mathcal{P}_t$ were drawn i.i.d. from some meta-distribution $\mathcal{Q}$ and we used $\gamma = \frac{120G}{\alpha} \max\left\{\frac{\sqrt{d\log\frac{1}{\delta}}}{\varepsilon m}, \frac{1}{G\sqrt{m}}\right\}$. Suppose we draw another task-distribution $\mathcal{P} \sim \mathcal{Q}$ with population risk $\ell_{\mathcal{P}}$ and minimizer $\theta_{\mathcal{P}}$, set $\hat{\phi} = \frac{1}{T}\sum_{t=1}^{T}\phi_t$, and run OGD with learning rate $\eta = \frac{V + \frac{1}{\alpha\sqrt{m}}}{\sqrt{m}}$ starting from $\hat{\phi}$ on $m$ samples from $\mathcal{P}$. Then the average iterate $\hat{\theta}$ satisfies*

$$\mathbb{E}(\ell_{\mathcal{P}}(\hat{\theta}) - \ell_{\mathcal{P}}(\theta^*)) \leq \frac{7GV}{2\sqrt{m}} + \frac{7201G}{\alpha}\max\left\{\frac{d\log\frac{1}{\delta}}{\varepsilon^2 m^2}, \frac{1}{m}\right\} + \frac{\alpha GD^2}{2T}(1 + \log T)$$

*for $V^2 = \min_{\phi \in \Theta}\mathbb{E}_{\mathcal{P} \sim \mathcal{Q}}\max_{\theta_{\mathcal{P}}}\|\phi - \theta_{\mathcal{P}}\|_2^2$.*

*Proof.* Applying online-to-batch conversion (e.g. Proposition A.1 in Khodak et al. (2019b)) twice and substituting Lemma A.2 yields

$$\mathbb{E}(\ell_{\mathcal{P}}(\hat{\theta}) - \ell_{\mathcal{P}}(\theta^*))$$

$$\leq \mathbb{E}\frac{\|\hat{\phi} - \theta^*\|_2^2}{2\eta m} + \eta G^2$$

$$\leq \mathbb{E}\frac{\|\phi^* - \theta^*\|_2^2}{2\eta m} + \eta G^2 + \frac{1}{2\eta m T}\sum_{t=1}^{T}\mathbb{E}\|\phi_t - \theta_t^*\|_2^2$$

$$\leq \mathbb{E}\frac{\|\phi^* - \theta^*\|_2^2}{2\eta m} + \eta G^2 + \frac{D^2\frac{1+\log T}{T} + 4\mathbb{E}\bar{V}^2}{2\eta m} + \frac{7200G^2}{\alpha^2\eta m}\max\left\{\frac{d\log\frac{1}{\delta}}{\varepsilon^2 m^2}, \frac{1}{m}\right\}$$

$$= \frac{5V^2}{2\eta m} + \eta G^2 + \frac{7200G^2}{\alpha^2\eta m}\max\left\{\frac{d\log\frac{1}{\delta}}{\varepsilon^2 m^2}, \frac{1}{m}\right\} + \frac{D^2}{2\eta m T}(1 + \log T)$$

where we have applied $\mathbb{E}\bar{V}^2 \leq V^2$. Substituting $\eta = \frac{V + \frac{1}{\alpha\sqrt{m}}}{G\sqrt{m}}$ yields the result. $\qquad\square$

## B  Experiment Details

**Datasets:**  We train a next word predictor for two federated datasets: (1) The Shakespeare dataset as preprocessed by (Caldas et al., 2018), and (2) a dataset constructed from Wikipedia articles, where each article is used as a different task. For each dataset, we set a fixed number of tokens per task, discard tasks with less tokens than the specified, and discard samples from those tasks with more. For Shakespeare, we set the number of tokens per task to 800 tokens, leaving 279 tasks for meta-training, 31 for meta-validation, and 35 for meta-testing. For Wikipedia, we set the number of tokens to $1,600$, which corresponds to having $2,179$ tasks for meta-training, 243 for meta-validation, and 606 for meta-testing. For the meta-validation and meta-test tasks, 75% of the tokens are used for local training, and the remaining 25% for local testing.

For the few-shot image classification experiments, we follow the standard set-up by splitting labels into training and testing and forming training tasks by randomly drawing labels from the training set. At evaluation time, we draw from the test set.

**Model Structure:**  Our model first maps each token to an embedding of dimension 200 before passing it through an LSTM of two layers of 200 units each. The LSTM emits an output embedding, which is scored against all items of the vocabulary via dot product followed by a softmax. We build the vocabulary from the tokens in the meta-training set and fix its length to $10,000$. We use a sequence length of 10 for the LSTM and, just as (McMahan et al., 2018), we evaluate using `AccuracyTop1` (i.e., we only consider the predicted word to which the model assigned the highest probability) and consider all predictions of the unknown token as incorrect. For Omniglot and Mini-ImageNet, we use the architectures from Nichol et al. (2018) to also match the ones from Finn et al. (2017). We evaluate in the standard transductive setting.

**Hyperparameters:**  For the language-modeling experiments, we tune the hyperparameters on the set of meta-validation tasks. For both datasets and all versions of the meta-learning algorithm, we tune hyperparameters in a two step process. We first tune all the parameters that are not related to refinement: the meta learning rate, the local (within-task) meta-training learning rate, the maximum gradient norm, and the decay constant. Then, we use the configuration with the best accuracy pre-refinement and then tune the refinement parameters: the refine learning rate, refine batch size, and refine epochs.

All other hyperparameters are kept fixed for the sake of comparison: full batch steps were taken on within-task data, with the maximum number of microbatches used for the task-global DP model. The parameter search spaces from which we sample are given in Tables 2, 3, 4 while Tables 5 and 6 contain our final choices. Note that the space for Local DP, especially in terms of the clipping threshold, is distinctively different from the others, as we did not find that searching through ranges similar to those for non-private and task-global DP led to learning high-quality meta-initializations.

For Omniglot, we largely based our hyperparameters on the choices of Nichol et al. (2018) for 5-way classification. We vary $m$, the number of training shots, but we continue to take 5 SGD steps of expected size $m$ within task and we leave the test-time SGD procedure exactly the same. However, we do tune for privacy clipping thresholds $\{0.01, 0.025, 0.05, 0.1, 0.2, 0.3, 0.4, 0.5\}$, Adam Learning Rates for meta-training tasks $\{10^{-4}, 5 \times 10^{-4}, 10^{-3}, 5 \times 10^{-3}\}$, and meta-batch sizes of $\{5, 15, 25, 50\}$.

For Mini-ImageNet, we perform a similar search except we also double the inner batch size to $2m$ (trading off less privacy amplification due to subsampling). We continue to tune for privacy clipping thresholds $\{0.01, 0.025, 0.05, 0.1, 0.3, 0.5, 0.7, 0.9, 1.1, 1.3, 1.5\}$, Adam Learning Rates $\{10^{-4}, 5 \times 10^{-4}, 10^{-3}, 5 \times 10^{-3}\}$, and meta-batch sizes of $\{5, 15, 25, 50\}$.

Table 2: Hyperparameter Search Space for Non-Private Training

|  | Shakespeare-800 | Wiki-1600 |
|---|---|---|
| Visits Per Task | $\{1,2,3,4,5,6,7,8,9\}$ | $\{1,2,3\}$ |
| Tasks Per Round | $\{5,10\}$ | $\{5,10\}$ |
| Within-Task Steps | $\{1,3,5,7,9\}$ | $\{1,3,5,7,9\}$ |
| Meta LR | $\{1,\sqrt{2},2,2\sqrt{2},4,4\sqrt{2},8,8\sqrt{2}\}$ | $\{1,\sqrt{2},2,2\sqrt{2},4,4\sqrt{2},8,8\sqrt{2}\}$ |
| Meta Decay Rate | $\{0,0.001,0.005,0.01,0.025,0.05,0.1\}$ | $\{0,0.001,0.005,0.01,0.025,0.05\}$ |
| Within-Task LR | $\{1,\sqrt{2},2,2\sqrt{2},4,4\sqrt{2},8\}$ | $\{1,\sqrt{2},2,2\sqrt{2},4,4\sqrt{2},8\}$ |
| $L_2$ Clipping | $\{0.4,0.5,0.6,0.7,0.8,0.9,1.0\}$ | $\{0.3,0.5,0.6,0.7,0.8,0.9,1.0\}$ |
| Refine LR | $\{0.1,0.15,0.3,0.5,0.7,0.8\}$ | $\{0.1,0.15,0.3,0.5,0.7,0.8\}$ |
| Refine Batch Size | $\{10,20,30,60\}$ | $\{10,20,30,60,120\}$ |
| Refine Epochs | $\{1,2,3\}$ | $\{1,2,3\}$ |

Table 3: Hyperparameter Search Space for Task-Global DP Training

|  | Shakespeare-800 | Wiki-1600 |
|---|---|---|
| Visits Per Task | $\{1,2,3\}$ | $\{1,2\}$ |
| Tasks Per Round | $\{5,10\}$ | $\{5,10\}$ |
| Within-Task Steps | 1 | 1 |
| Meta LR | $\{1,\sqrt{2},2,2\sqrt{2},4,4\sqrt{2},8,8\sqrt{2}\}$ | $\{1,\sqrt{2},2,2\sqrt{2},4,4\sqrt{2},8,8\sqrt{2}\}$ |
| Meta Decay Rate | $\{0,0.001,0.005,0.01,0.025,0.05,0.1\}$ | $\{0,0.001,0.005,0.01,0.025,0.05\}$ |
| Within-Task LR | $\{1,\sqrt{2},2,2\sqrt{2},4,4\sqrt{2},8\}$ | $\{1,\sqrt{2},2,2\sqrt{2},4,4\sqrt{2},8\}$ |
| $L_2$ Clipping | $\{0.4,0.5,0.6,0.7,0.8,0.9,1.0\}$ | $\{0.3,0.4,0.5,0.6,0.7,0.8,0.9,1.0\}$ |
| Refine LR | $\{0.1,0.15,0.3,0.5,0.7,0.8\}$ | $\{0.1,0.15,0.3,0.5,0.7,0.8\}$ |
| Refine Batch Size | $\{10,20,30,60\}$ | $\{10,20,30,60,120\}$ |
| Refine Epochs | $\{1,2,3\}$ | $\{1,2,3\}$ |

Table 4: Hyperparameter Search Space for Local-DP Training

|  | Shakespeare-800 | Wiki-1600 |
|---|---|---|
| Visits Per Task | $\{1,2,3\}$ | $\{1,2\}$ |
| Tasks Per Round | $\{5,10,20\}$ | $\{10,20,40,80\}$ |
| Within-Task Steps | $\{1,2,3\}$ | $\{1,2,3\}$ |
| Meta LR | $\{1,\sqrt{2},2,2\sqrt{2},4,4\sqrt{2},8,8\sqrt{2}\}$ | $\{1,\sqrt{2},2,2\sqrt{2},4,4\sqrt{2},8,8\sqrt{2}\}$ |
| Meta Decay Rate | $\{0,0.001,0.005,0.01,0.025,0.05,0.1\}$ | $\{0,0.001,0.005,0.01,0.025,0.05\}$ |
| Within-Task LR | $\{1,\sqrt{2},2\sqrt{2},4,4\sqrt{2},8\}$ | $\{1,\sqrt{2},2\sqrt{2},4,4\sqrt{2},8\}$ |
| $L_2$ Clipping | $\{0.005,0.01,0.025,0.05,0.1,0.25,0.5\}$ | $\{0.005,0.01,0.025,0.05,0.1,0.25\}$ |
| Refine LR | $\{0.1,0.15,0.3,0.5,0.7,0.8\}$ | $\{0.1,0.15,0.3,0.5,0.7,0.8\}$ |
| Refine Batch Size | $\{10,20,30,60\}$ | $\{10,20,30,60,120\}$ |
| Refine Epochs | $\{1,2,3\}$ | $\{1,2,3\}$ |

Table 5: Final Hyperparameters for Shakespeare-800

|  | Non-private | T-G $\varepsilon = 22.5$ | T-G $\varepsilon = 9.2$ | T-G $\varepsilon = 4.5$ | Local $\varepsilon = 22.5$ | Local $\varepsilon = 9.2$ | Local $\varepsilon = 4.5$ |
|---|---|---|---|---|---|---|---|
| Visits Per Task | 7 | 2 | 2 | 1 | 2 | 2 | 1 |
| Tasks Per Round | 5 | 5 | 5 | 5 | 20 | 5 | 20 |
| Within-Task Steps | 5 | 1 | 1 | 1 | 4 | 1 | 2 |
| Meta LR | 8 | $8\sqrt{2}$ | 8 | $8\sqrt{2}$ | $4\sqrt{2}$ | $4\sqrt{2}$ | 4 |
| Meta Decay Rate | 0.01 | 0.01 | 0.01 | 0.05 | 0.1 | 0 | 0 |
| Within-Task LR | 2 | $2\sqrt{2}$ | 2 | 2 | $4\sqrt{2}$ | $4\sqrt{2}$ | 1 |
| $L_2$ Clipping | 0.5 | 0.6 | 0.5 | 0.4 | 0.1 | 0.01 | 0.01 |
| Refine LR | 0.15 | 0.5 | 0.1 | 0.3 | 0.8 | 0.8 | 0.5 |
| Refine Batch Size | 30 | 10 | 60 | 30 | 10 | 10 | 10 |
| Refine Epochs | 1 | 1 | 1 | 3 | 3 | 3 | 3 |

Table 6: Final Hyperparameters for Wiki-800

|  | Non-private | T-G $\varepsilon = 22.5$ | T-G $\varepsilon = 9.2$ | T-G $\varepsilon = 4.5$ | Local $\varepsilon = 22.5$ | Local $\varepsilon = 9.2$ | Local $\varepsilon = 4.5$ |
|---|---|---|---|---|---|---|---|
| Visits Per Task | 2 | 1 | 1 | 1 | 1 | 1 | 1 |
| Tasks Per Round | 5 | 10 | 10 | 20 | 20 | 20 | 20 |
| Within-Task Steps | 3 | 1 | 1 | 1 | 2 | 2 | 2 |
| Meta LR | $2\sqrt{2}$ | $2\sqrt{2}$ | 4 | 4 | 8 | $4\sqrt{2}$ | 8 |
| Meta Decay Rate | 0.001 | 0 | 0.001 | 0.005 | 0.005 | 0.025 | 0 |
| Within-Task LR | 2 | 8 | $4\sqrt{2}$ | 8 | $2\sqrt{2}$ | $2\sqrt{2}$ | $2\sqrt{2}$ |
| $L_2$ Clipping | 1 | 0.8 | 0.7 | 0.8 | 0.025 | 0.05 | 0.005 |
| Refine LR | 0.1 | 0.8 | 0.5 | 0.7 | 0.8 | 0.8 | 0.8 |
| Refine Batch Size | 10 | 10 | 60 | 10 | 10 | 10 | 10 |
| Refine Epochs | 1 | 2 | 2 | 2 | 2 | 2 | 3 |

