# OpenReview forum: "Differentially Private Meta-Learning"
_ICLR.cc/2020/Conference — Accept (Poster)_

### Official Review · AnonReviewer2 · 2019-10-23
**Official Blind Review #2**

**Rating:** 6

**Review:**

This paper proposes the notions of different privacy levels for different attack models, namely global and local meta-level and within-task level privacy for meta-learning. It proposes an algorithm for global within-task privacy. It provides privacy and utility guarantee of the proposed algorithm and experimental evaluations.

The proposed definitions make sense to me for the scenarios mentioned in the paper. The utility guarantee also seems interesting. I’m a little concerned with the significance and novelty of the proposed algorithm (it seems like a direct application of a generic DPSGD algorithm) and the utility analysis. Maybe you can justify more on that part. I think the experimental evaluation can be made more complete, for example, you may consider:
- a convex setting as was considered in the utility guarantee,
- varying epsilon values. I think a utility vs. epsilon curve can better support your paper.

**Experience Assessment:**

I have read many papers in this area.

**Review Assessment: Checking Correctness Of Derivations And Theory:**

I assessed the sensibility of the derivations and theory.

**Review Assessment: Checking Correctness Of Experiments:**

I assessed the sensibility of the experiments.

**Review Assessment: Thoroughness In Paper Reading:**

I read the paper at least twice and used my best judgement in assessing the paper.

---

> ### Author Response · Authors · 2019-11-14
> **Response To Review #2**
>
> Thank you for your time and thoughtful review! We hope to address your comments below. [note: We decided to respond to each reviewer individually, though we note that there is significant overlap in our responses to R1 and R2 since these two reviewers had several similar comments/suggestions]
>
> 1) Varying epsilon:
> -Thank you for the suggestion - we agree that observing performance across different privacy budgets is a valuable experiment. Since the submission deadline, we have run each of the language-modelling experiments for two additional privacy budgets. These epsilons, 22.5 and 4.5, were determined by doubling and halving the noise multiplier for the Local DP baseline. We observe, as expected, that higher utility is traded off with lower privacy and vice versa. However, we also see that even when stricter privacy is required (epsilon = 4.5), fairly useful/competitive meta-learned initializations are still obtainable (see the updated Figure 2).
>
> 2) Convex experiments:
> -We also acknowledge convex case results as a valuable addition to our experiments. Though we currently do not have updates for these experiments we fully intend to include such results for future revision. Specifically, we have been looking at adding evaluations with linear models on the datasets used by Denevi et al. (2019) and Khodak et al. (2019).
>
> 3) Novelty of techniques:
> -We recognize that our method is quite simple and takes advantage of existing single-task privacy methods. However, we see this simplicity as a benefit of our approach, in the same way that non-private methods such as MAML and Reptile are simple extensions of regular single-task gradient descent. This simplicity enables flexible application to a variety of models and settings.
>
> 4) Significance of utility guarantee:
> -We believe the utility guarantee is significant for several reasons: (1) it is the first such result in the meta-learning setting and represents a reasonable tradeoff between privacy and accuracy when compared with existing (non-private) theory for meta-learning (Denevi et al., 2019; Khodak et al., 2019); (2) existing results for the federated learning setting often focus on how much noise a privacy mechanism adds rather than the effect on statistical performance (Agarwal et al., 2018; Bhowmick et al., 2019; Truex et al., 2019), which is what we care about directly; (3) while the analysis is convex-case, simple linear models may be of interest in application areas of private meta-learning such as learning from electronic medical records.
>
> References:
> -Agarwal, Suresh, Yu, Kumar, McMahan. cpSGD: Communication-efficient and differentially-private distributed sgd. NeurIPS 2018.
> -Bhowmick, Duchi, Freudiger, Kapoor, Rogers. Protection against reconstruction and its applications in private federated learning. 2019. https://arxiv.org/abs/1812.00984
> -Denevi, Ciliberto, Grazzi, Pontil. Learning-to-learn stochastic gradient descent with biased regularization. ICML 2019.
> -Khodak, Balcan, Talwalkar. Provable guarantees for gradient-based meta-learning. ICML 2019.
> -Truex, Baracaldo, Anwar, Steinke, Ludwig, Zhang. A hybrid approach to privacy-preserving federated learning. 2019. https://arxiv.org/abs/1812.03224

---

### Official Review · AnonReviewer3 · 2019-11-04
**Official Blind Review #3**

**Rating:** 6

**Review:**

As a non-expert in differential privacy, my review is based on my educated guess and limited understanding of the paper.

This paper considers a the differential privacy problem regarding the parameter-tranfer algorihtm in meta-learning, such as MAML and Repile. To me, the setting is very interesting and according to the paper, it seems that it is the first formalization for this setting. Since meta-learning is becoming more and more popular, the paper possesses practical values for privacy-preseving meta-learning algorithms.

The proposed differential privacy seetings are well illustrated and presented in the paper. The differentially private parameter-transfer is also straightforward but has been twisted a little bit for theoretical guarantees. The theoretical results seem pretty reasonable to me, but I have not checked the proof in detail.

The experiments demonstrate the effectiveness of the proposed differentially private parameter-transfer. However, the experiments are also very toy-ish in some senses, which reduces the the pritical values. All the datasets the paper uses are very easy ones. It is highly recommended to perform experiments on more challenging meta-learning datasets, such as Mini-ImageNet, CUB, etc. See [A Closer Look at Few-shot Classification, ICLR 2019] as an example for conducting few-shot learning experiments.

**Experience Assessment:**

I do not know much about this area.

**Review Assessment: Checking Correctness Of Derivations And Theory:**

I did not assess the derivations or theory.

**Review Assessment: Checking Correctness Of Experiments:**

I assessed the sensibility of the experiments.

**Review Assessment: Thoroughness In Paper Reading:**

I made a quick assessment of this paper.

---

> ### Author Response · Authors · 2019-11-14
> **Response to Review #3**
>
> Thank you for your time and thoughtful review! We agree that the use of more a challenging dataset would bolster the practical value of our work. Since the deadline, we have run a set of experiments for 5-shot-5-way Mini-ImageNet similar to the ones we already had for 5-shot-5-way Omniglot (see the updated Figure 3). Qualitatively, the results and their conclusions are fairly similar as well – while a local DP approach cannot even improve upon single-task learning, our method allows individual tasks to benefit from multi-task data via our task-global notion of DP.

---

### Official Review · AnonReviewer4 · 2019-11-08
**Official Blind Review #1**

**Rating:** 6

**Review:**

This paper considers the problem of achieving formal privacy guarantees in the context of parameter-sharing meta learning. This is a problem that has been studied recently, although generally not under the exact record-level task-privacy model studied in this paper (what they call global task privacy). The problem is well-motivated: since in meta learning we want to leverage information from similar tasks to increate data efficiency, there may be privacy concerns for each of the task owners about both other task owners and the aggregator (meta-learner).


Their approach is conceptually simple; they use standard private stochastic gradient descent within each task to provide task level privacy. In combination with post-processing and composition guarantees this gives privacy for the overall mechanism. They are able to show theoretical guarantees via the standard accuracy guarantees of private SGD and no-regret
guarantees of OCO. They show a bound on the expected transfer risk:
O(V/sqrt(m) + 1/Tsqrt(m)) + o(1/sqrt(m)) which is close to the non-private bound from Denevi (2019).

They evaluate the empirical performance of these models via a transfer learning setting  where they are training a deep RNN for next word prediction on two large corpi, and the tasks correspond to individual articles. They show that the private variate of Reptile is competitive with the non-private variant in these settings.
A few comments
-	It is odd to just fix epsilon = 9.2 instead of showing a Pareto curve. Why this particular value?
-	Simpler (convex) experiments to illustrate the theoretical guarantees would improve the paper

Overall I like the motivation and the theory results are solid, if not a bit obvious. However, due to the lack of novelty in any of the applied techniques, and the fact that the experiments could be expanded, I recommend a weak accept.

**Experience Assessment:**

I have published in this field for several years.

**Review Assessment: Checking Correctness Of Derivations And Theory:**

I assessed the sensibility of the derivations and theory.

**Review Assessment: Checking Correctness Of Experiments:**

I assessed the sensibility of the experiments.

**Review Assessment: Thoroughness In Paper Reading:**

I made a quick assessment of this paper.

---

> ### Author Response · Authors · 2019-11-14
> **Response to Review #1**
>
> Thank you for your time and thoughtful review! We hope to address your comments below. [note: We decided to respond to each reviewer individually, though we note that there is significant overlap in our responses to R1 and R2 since these two reviewers had several similar comments/suggestions]
>
> 1) Varying epsilon:
> -Thank you for the suggestion - we agree that observing performance across different privacy budgets is a valuable experiment. Since the submission deadline, we have run each of the language-modelling experiments for two additional privacy budgets. The original choice of 9.2 was somewhat arbitrary, as emphasis was placed on showing that meta-learning could be compatible with a “reasonable” level of privacy (using our notion of privacy). However, we hope by including these additional results, more context is provided in terms of the accuracy-privacy trade-off. The new epsilons used, 22.5 and 4.5, were determined by doubling and halving the noise multiplier for the Local DP baseline. We observe, as expected, that higher utility is traded off with lower privacy and vice versa. However, we also see that even when stricter privacy is required (epsilon = 4.5), fairly useful/competitive meta-learned initializations are still obtainable (see the updated Figure 2).
>
> 2) Convex experiments:
> -We also acknowledge convex case results as a valuable addition to our experiments. Though we currently do not have updates for these experiments we fully intend to include such results for future revision. Specifically, we have been looking at adding evaluations with linear models on the datasets used by Denevi et al. (2019) and Khodak et al. (2019).
>
> 3) Novelty of techniques:
> -We recognize that our method is quite simple and takes advantage of existing single-task privacy methods. However, we see this simplicity as a benefit of our approach, in the same way that non-private methods such as MAML and Reptile are simple extensions of regular single-task gradient descent. This simplicity enables flexible application to a variety of models and settings.
>
> References:
> -Denevi, Ciliberto, Grazzi, Pontil. Learning-to-learn stochastic gradient descent with biased regularization. ICML 2019.
> -Khodak, Balcan, Talwalkar. Provable guarantees for gradient-based meta-learning. ICML 2019.

---

### Author Response · Authors · 2019-11-14
**Updated Revision**

We thank the reviewers for their careful reading of the paper and their useful comments and responses. We have updated our submission to incorporate the feedback given. In particular, we were able to add experiments examining the performance of our approach on next-word prediction across different privacy budgets, as suggested by R1 and R2, as well as experiments showing the performance of our method on few-shot learning using the more-challenging Mini-ImageNet dataset, as suggested by R3. On both sets of experiments, we continue to see that our approach is able to exploit our notion of DP to out-perform the main competing approach.

Here we list the major additions as well as some important minor changes/errata.

Major Additions:
-Experimental results for different privacy budgets were added as additional learning curves to Figure 2.
-Experimental results for Mini-ImageNet were added as additional plots to Figure 3.

Minor Changes:
-The graphs in Figure 3 have been corrected to read m=10, m=20, m=30 instead of all being m=30. We also added the privacy budget of epsilon=9.5 to the graph.
-Formatting for Figures 2, 3 such as legend placement and color scheme were adjusted for clarity.

---

### Decision · Program_Chairs · 2019-12-19

**Decision:**

Accept (Poster)

**Comment:**

Thanks to the authors for the submission. This paper studies differentially private meta-learning, where the algorithm needs to use information across several learning tasks to protect the privacy of the data set from each task. The reviewers agree that this is a natural problem and the paper presents a solution that is essentially an adoption of differentially private SGD. There are several places the paper can improve. For the experimental evaluation, the authors should include a wider range of epsilon values in order to investigate the accuracy-privacy trade-off. The authors should also consider expanding the existing experiments with other datasets.